# Integration of Multiple Interferometers in Highly Multiplexed Diagnostic KITs to Evaluate Several Biomarkers of COVID-19 in Serum

**DOI:** 10.3390/bios12090671

**Published:** 2022-08-23

**Authors:** Ana María M. Murillo, Luis G. Valle, Yolanda Ramírez, María Jesús Sánchez, Beatriz Santamaría, E. Molina-Roldan, Isabel Ortega-Madueño, Elena Urcelay, Luca Tramarin, Pedro Herreros, Araceli Díaz-Perales, María Garrido-Arandia, Jaime Tome-Amat, Guadalupe Hernández-Ramírez, Rocío L. Espinosa, María F. Laguna, Miguel Holgado

**Affiliations:** 1Center for Biomedical Technology (CTB), Universidad Politécnica de Madrid, Parque Científico y Tecnológico de la UPM, Campus de Montegancedo, Pozuelo de Alarcón, 28223 Madrid, Spain; 2Health Research Institute of the Hospital Clínico San Carlos, IdISSC. C/Profesor Martín Lagos s/n, 4ª Planta Sur, 28040 Madrid, Spain; 3BioOptical Detection S.L., Centro de Empresas, Campus Montegancedo, 28223 Madrid, Spain; 4Escuela Técnica Superior de Ingenieros Industriales, Universidad Politécnica de Madrid, C/José Gutiérrez Abascal 2, 28006 Madrid, Spain; 5Escuela Técnica Superior de Ingeniería y Diseño Industrial, Universidad Politécnica de Madrid, Ronda de Valencia 3, 28012 Madrid, Spain; 6Center for Plant Biotechnology and Genomics (CBGP), Universidad Politécnica de Madrid, Parque Científico y Tecnológico de la UPM, Campus de Montegancedo, Pozuelo de Alarcón, 28223 Madrid, Spain

**Keywords:** COVID-19 biomarkers, SARS-CoV-2, serum, immunoglobulin, ferritin, interferometric optical detection method

## Abstract

In the present work, highly multiplexed diagnostic KITs based on an Interferometric Optical Detection Method (IODM) were developed to evaluate six Coronavirus Disease 2019 (COVID-19)-related biomarkers. These biomarkers of COVID-19 were evaluated in 74 serum samples from severe, moderate, and mild patients with positive polymerase chain reaction (PCR), collected at the end of March 2020 in the Hospital Clínico San Carlos, in Madrid (Spain). The developed multiplexed diagnostic KITs were biofunctionalized to simultaneously measure different types of specific biomarkers involved in COVID-19. Thus, the serum samples were investigated by measuring the total specific Immunoglobulins (sIgT), specific Immunoglobulins G (sIgG), specific Immunoglobulins M (sIgM), specific Immunoglobulins A (sIgA), all of them against SARS-CoV-2, together with two biomarkers involved in inflammatory disorders, Ferritin (FER) and C Reactive Protein (CRP). To assess the results, a Multiple Linear Regression Model (MLRM) was carried out to study the influence of IgGs, IgMs, IgAs, FER, and CRP against the total sIgTs in these serum samples with a goodness of fit of 73.01% (Adjusted R-Squared).

## 1. Introduction

The Severe Acute Respiratory Syndrome Coronavirus 2 (SARS-CoV-2) is responsible for a worldwide pandemic causing millions of infected people and a significant number of deaths [1,2,3]. In Spain, the Coronavirus Disease 2019 (COVID-19) pandemic caused by SARS-CoV-2 started at the beginning of 2020, with Madrid (Spain) being one of the areas with a higher number of cases.

SARS-CoV-2 is a single-stranded RNA-enveloped virus whose genes S, E, M, and N encode structural proteins. Unlike other functional proteins of SARS-CoV-2, the S protein on the viral surface is responsible for virus entry into the host cells, is able to induce the host immune response, and is the main antigen of the virus to elicit neutralizing antibodies [1,4,5]. However, it must be considered that studies carried out in Singapore populations and by the Wuhan Union Hospital indicate that a percentage of 0.6 to 12% of patients with COVID-19 do not produce specific antibodies against SARS-CoV-2 [6,7], and a percentage of children are not affected by the virus, as reported in several works; for example, a study across 25 European countries showed 16% of asymptomatic cases to be in children [8,9,10].

In the study carried out at the University Hospital of La Paz (Madrid, Spain), the severity of the COVID-19 disease was evaluated on a sample of more than 2226 patients (this cohort of patients included patients hospitalized from 25 February to 19 April 2020), with a higher incidence in the elderly (the median age reported in the article was 61 years old). Global mortality was over 20% (460 of the 2226 patients died), with a higher percentage in men. The mortality rate surpassed 40% in men over 70 y/o and in women over 80 y/o. For elders older than 80 y/o, this mortality rate increased to above 50% on average [11].

The Polymerase Chain Reaction technique (PCR) is the accepted gold-standard diagnostic method for molecular detection of SARS-CoV-2 to control the pandemic crisis [5]. However, a great need remains for assays that measure immunity antibody responses, determine seroconversion, measure other related biomarkers of COVID-19, and study the immune response to SARS-CoV-2 [3,5,12]. Relevant technologies have been published regarding the detection of specific antibodies associated with the immune response of the COVID-19 infection, such as those based on enzyme-linked immunosorbent assay (ELISA), chemiluminescence (CLIA), or lateral flow (LF), among other alternatives. In addition, tests based on non-invasive samples such as saliva also facilitate the population-based mass screening of COVID-19, in an attempt to overcome the current pandemic situation [3,12,13].

Even though the viral load has been revealed to be a significant factor for the prognosis and monitoring of the disease, relevant studies suggest that the death of patients is caused by a defensive uncontrolled reaction of our immune system, a cytokine storm, and not by the virus itself. In this sense, it should be noted that high levels of interleukins such as IL-6 or TNFa have been reported in peripheral blood samples and have allowed medical practitioners to apply promising therapies to block their effect, inhibiting proinflammatory pathways with different drugs like Tocilizumab or Fedratinib [14,15,16].

Although PCR is a technique with high specificity, it is also a laborious technique, with long waiting times and the requirement of difficult sample processing to extract the genetic material in order to amplify it. On the other hand, simpler diagnostic tests such as those based on LF have less sensitivity for low concentrations of the target biomarker. This makes necessary the implementation of new diagnostic systems that allow high sensitivity and specificity but with simpler handling, low-cost equipment, and faster analysis.

For this reason, a wide variety of biosensors have been developed, including field-effect transistor (FET)-based biosensors [17], electrochemical biosensors [18,19], and surface plasmon resonance (SPR)-based biosensors [20].

In a recent previous work, we reported an immunosensor for measuring specific immunoglobulins in sera and saliva [21]. In the present work, we increased the demultiplexing capacity to be able to measure six biomarkers in a single KIT: sIgT, sIgG, sIgM, sIgA, FER, and CRP. We developed an in vitro diagnostic system based on a highly multiplexed KIT with 65 interferometric biotransducers in a single KIT. The basis of the interferometric optical-based technology employed to perform this qualitative and quantitative study has been described previously in detail, and it is well correlated with ELISA [21,22]. These biotransducers, also called Biophotonic Sensing Cells (BICELLs) are well reported and described in the literature [23,24,25]. The in vitro detection system is based on the Interferometric Optical Detection Method (IODM) [26,27], whose readout signal is measured in ΔIROP (%) (Increased Relative Optical Power) to analyze six different biomarkers per sample in a rapid, cost-effective, affordable, and reliable manner. In this work, we report the significant changes necessary to overcome the challenge of measuring multiple biomarkers in a single diagnostic KIT.

The present work reports the analysis of 74 serum samples of severe (25), moderate (26), and mild (23) patients with a positive PCR, collected at the end of March 2020 from the Hospital Clínico San Carlos in Madrid (Spain). Patients were divided into these severity groups based on their hospitalization in the intensive care unit and the requirement for mechanical ventilation. Mild patients were those who did not require hospitalization, moderate patients were defined as those who were hospitalized but did not require admission to the ICU and mechanical ventilation, and finally, severe patients were those who required admission to the ICU and mechanical ventilation.

The aim of this study was to evaluate the immune response by measuring the titer of total specific Immunoglobulins (sIgT), specific Immunoglobulins G (sIgG), specific Immunoglobulins M (sIgM), and specific Immunoglobulins A (sIgA) against SARS-CoV-2, together with the inflammatory biomarkers Ferritin (FER) and C Reactive Protein (CRP), to evidence differences associated with the immunoresponse.

## 2. Materials and Methods

### 2.1. Reagents and Chemicals

Anti-Ferritin heavy chain (α-FTH1) and anti-C Reactive Protein (α-CRP) antibodies, Bovine Serum Albumin (BSA), casein hydrolysate, Phosphate Buffered Saline (PBS), and secondary antibodies (α-IgG, α-IgM and α-IgA) were purchased from Sigma-Aldrich, St. Louis, MO, USA.

### 2.2. Samples and Patients

Seventy-four serum samples of patients with SARS-CoV-2 infection confirmed by PCR and twenty samples from blood donors were provided by the biobank of the Hospital Clínico San Carlos (HCSC) (B.0000725; PT17/0015/0040; ISCIII-FEDER). These samples were characterized with information about symptoms and blood groups complying with current legal ethical regulations, with the collaboration of the Instituto de Medicina de Laboratorio (IML) and the Unidad de Inovación of the HCSC. A total of 20 of these samples belonged to asymptomatic patients.

Once these samples were received, they were heated at 56 °C to inactivate the complement and to reduce the potential risk from any residual virus. Then, serum was diluted at 1:10 and stored at −80 °C until used.

### 2.3. Recombinant SARS-CoV-2 S1 Protein

To produce the recombinant virus protein, SARS-CoV-2 cDNA was kindly donated by Isabel Solá (Centro Nacional de Biotecnología, Spanish National Research Council (CNB-CSIC, Spain)). This recombinant SARS-CoV-2 spike protein (rS1) was produced in Pichia Pastoris and purified using the methods described in [21,28].

### 2.4. Fabrication of Multiplexed Diagnostic KITs

A more detailed description of the fabrication of KITs by the Optics, Photonics, and Biophotonics Group can be found in previous works [22,23,26]. We designed the diagnostic KIT with 65 wells with a diameter of 1 mm, made of Polyvinyl Chloride (PVC), with a Fabry-Perot interferometric biosensing site with a diameter of 200 microns, made of SU-8 epoxy resist within each well, which only needs a drop with a sample volume of 1 µL (Figure 1). The interferometric signal produced by each biosensing site was read out vertically using an optical reader. In summary, a negative resist (SU-8) was spin-coated over a Silicon wafer, then a photolithography process was performed to create a multiplexed design with 65 sensing cells over the surface, as previously described [21]. After a dicing step, the multiplexed KITs were finalized by fixing the wafer chips to glass slides. Figure 2 represents the design of this highly multiplexed diagnostic KIT and the assays carried out.

### 2.5. Biofunctionalization of the Multiplexed Diagnostic KITs

First, the surface of the biosensing sites was activated through an O2 plasma [21,29] process to immobilize the specific bioreceptors, rS1, α-FTH, and α-CRP, by covalent binding. The diagnostic kits were incubated with 300 ng of rS1 (1 μL at 300 μg/mL in PBS (pH 7.5)), 100 ng of α-FTH1 (1 μL at 100 μg/mL in PBS (pH 7.5)), and 100 ng of α-CRP (1 μL at 100 μg/mL in PBS (pH 7.5)) until saturation, using an automated liquid dispensing platform, BioDot AD1520TM. BSA was incubated as a negative control (1 μL at 50 μg/mL in PBS (pH 7.5)) to check the selectivity of the sensing system. After that, proteins were incubated in a humid environment at 37 °C for 3 h.

After the incubation time, the diagnostic KITs were washed with Milli-Q water and dried with particle filters and dry air. Then, the readout signal (ΔIROP (%)) of each sensing site was measured, and the values were confirmed to fit within our admissible limits of tolerance. To prevent non-specific binding on the remaining binding surface, all diagnostic KITs were blocked with casein hydrolysate 1× for 1 h under agitation.

The BioDot dispensing system is able to dispense accurately small volumes of reagent (on the order of 100 nanoliters), allowing excellent reproducibility in the biofunctionalization stage.

### 2.6. In Vitro Detection of IgT, IgG, IgM, IgA, FER, and CRP

First, the diagnostic kits were biofunctionalized with rS1 protein and α -FTH1 and α -CRP antibodies, in addition to BSA protein as a negative control, and then were blocked. After this process, SARS-CoV-2 specific antibodies, FER and CRP, were measured. Serum samples were diluted at 1:10 and incubated in order to measure sIgT, FER, and CRP directly in a single step (Figure 2). In this case, specific antibodies (IgG, IgM, and IgA) to the virus, ferritin, and PCR were measured in each KIT for two different patients (together with their negative controls), and the positive control (PCR-confirmed COVID-19 patient) was incubated in each diagnostic KIT. After the washing process, the FER, CRP, and sIgT levels of the patients were directly detected.

Titers of the specific antibodies were analyzed simultaneously. The concentrations of IgGs, IgMs, and IgAs were determined by secondary antibodies (α-IgG, α-IgM, and α-IgA) (Figure 2). The readout signal ΔIROP (%) was measured after each stage of incubation.

For each biomarker, we examined the difference in the patient signal corrected by the BSA signal in the corresponding diagnostic KIT to determine the specific signal we were measuring, since we eliminated the background signal produced by each patient’s serum. The signal obtained in a serum sample of a clinically tested healthy patient minus the BSA signal in the corresponding diagnostic KIT plus the standard uncertainty was established as a negative control [30]. As a positive control signal, we considered that resulting from the serum sample of a clinically diagnosed patient. Both signals were used to control the quality of the measurements during the trial.

As the sample signals were referenced to the BSA level, and 5 of a given serum sample signal is the difference between the signal in the corresponding biomarker (sIgT, sIgG, sIgM, sIgA, FER, and CRP), normalized to the BSA signal, this protocol ensures the comparability of the data collected for this trial.

### 2.7. Statistical Comparisons among the Severe, Moderate, and Mild Patients’ Groups

We analyzed whether the degree of severity (Severe, Moderate, and Mild groups, as classified by the hospital) was a significant factor for the variable response IgT (the total concertation of specific immunoglobulins against SARS-CoV-2). As a result, the ANalysis Of VAriance (ANOVA) showed that the degree of severity was significant for the variable response (IgT) with a *p*-value of 0.045. When we compared the IgT for each of the different groups of severity (Severe, Moderate, and Mild), we observed that the Severe group was different from the Moderate and Mild groups. The pairwise comparison showed that, indeed, Severe was different from Moderate and Mild, with *p*-values of 0.028 and 0.036 respectively. Therefore, in terms of IgT, the groups Moderate and Mild were different from the group of Severe patients. However, the Moderate and Mild groups did not present significant differences (*p*-value = 0.917) in terms of IgT.

### 2.8. Multiple Linear Regression Models

In this case, study we employed a Multiple Linear Regression Model (MLRM) to evaluate the titers of sIgG, sIgM, and sIgA FER and CRP, and how they were correlated with the sIgT.

The steps for obtaining this model were as follows: we firstly obtained the correlation matrix among all the quantitative variables (sIgT, sIgG, sIgM, sIgA, FER, and CRP). The correlation results are presented at the end of the document. Secondly, we obtained the simple regression models of sIgT against sIgG, sIgM, sIgA, FER, and CRP to test that the variables were significant (see Table 1). Finally, we evaluated all the variables in the Multiple Regression Model, noting that the variable CRP was not significant, with a *p*-value of 0.472, which was much higher than the considered statistical significance level of 0.05. Finally, we removed the CRP variable for the Multiple Regression Model obtaining a goodness of fit of 70.29% (Adjusted R-Squared).

Finally, we considered the qualitative variable of Severe, Moderate, and Mild in the Multiple Regression Model, allowing us to achieve a goodness of fit of 73.01% (Adjusted R-Squared). All the models presented a good statistical diagnosis.

## 3. Results

Four different proteins were immobilized as bioreceptors onto each interferometric biosensing site of the diagnostic KIT: α-FTH1, α-CRP, rS1, and BSA. In a first step, the patients’ samples were incubated, washed, dried, and optically readout to recognize FER, CRP, and sIgT, which mainly correspond to the compendium of sIgG, sIgM, sIgA against SARS-CoV-2. In a second step, we identified and classified the different types of antibodies incubating anti-human IgG (α-IgG), anti-human IgM (α-IgM), and anti-human IgA (α-IgA).

We measured the serum samples and analyzed the different biomarkers for each Level of Severity of COVID-19 (LSC) following the above-mentioned biosensing strategy. These results can be observed in serum samples from patients classified as severe (Figure 3), moderate (Figure 4), and mild (Figure 5). The level of the readout signal for the titer of sIgT, FER, and CRP is proportional to the concentration of those molecules in the sample [26,27]. However, the signal of the titers of sIgG, sIgM, and sIgA is proportional to the concentration of the secondary antibody employed to specifically recognize the human sIgG, sIgM, and sIgA present in the serum samples.

To compare these results, we calculated the percentage of positives, considering the six biomarkers for each LSC. The results can be observed in Figure 6.

We observed that 96% of Severe donors showed a positive sIgT signal, in contrast to 77% of Moderate and 65% of Mild patients.

Over 72% of Severe donors showed a positive sIgG signal, in contrast to over 54% of Moderate and 39% of Mild patients. Over 48% of Severe donors showed a positive sIgM signal, higher than in Moderate and Mild patients. Only 44% of Severe donors showed a positive sIgA signal, in contrast to 38 and 57% of Moderate and Mild patients, respectively.

Regarding the inflammatory biomarkers, over 60% of Severe patients showed significantly positive FER values in contrast to 58% of Moderate and 43% of Mild patients. There was not a trend in CRP positive values of Severe, Moderate, and Mild donors.

From a qualitative point of view, we observed that the positivity of the selected biomarkers depends on the severity of COVID-19, leading to a higher titer of antibodies, particularly for sIgT and sIgG for Severe patients, in contrast to the titer of sIgA. It is also observed that the percentage of positives in FER is higher for Severe patients, and no differences were found in the percentage of CRP.

We considered it worthy to analyze some asymptomatic cases. Twenty blood donors from the hospital Clínico San Carlos were analyzed in order to obtain a preliminary figure of potential asymptomatic cases. As a result, we included some cases at the end of February 2020, and we also measured some volunteers from our research center at the end of May 2020.

We discovered that some blood donors presented a significant titer of specific antibodies to SARS-CoV-2 (Figure 7), and although the sample of 20 donors was limited, the roughly five cases found to represent a percentage of 25%.

Finally, we studied these experimental results by analyzing the degree of correlation of each biomarker measured (sIgG, sIgM, sIgA, FER, and CRP) in the diagnostic KIT.

In this basic model, we observed that sIgG, sIgM, and FER were rather more significant than sIgAs and CRP. It could be assumed that sIgAs do not provide much more additional information once IgGs and IgMs have been analyzed, and CRP behaves as an unspecific biomarker for COVID-19. In Figure 8, this correlation matrix can be observed:

The *p*-values for each biomarker are presented in Table 1.

Once the individual correlation of each of the individual biomarkers with the total amount of specific antibodies against SARS-CoV-2 had been analyzed, we analyzed a multiple regression model correlating the sIgT with the biomarkers sIgG, sIgM, sIgA, FER, and CRP. In this model, we observed that CRP was not a significant variable, confirming that CRP is an unspecific biomarker due to its expression in multiple inflammatory pathologies [31]. In fact, the *p*-value of CRP in this model of 0.47 is much higher than the statistically significant level considered of 0.05. Thus, avoiding the CRP in the multiple regression model, we observed that sIgG (*p*-value = 1.5 × 10^−10^), sIgM (*p*-value = 6.08 × 10^−8^), sIgA (*p*-value = 7.09 × 10^−8^) and FER (*p*-value = 1.7 × 10^−4^) were highly significant. Considering this final model, we calculated the coefficients to explain the sIgT as a function of the abovementioned biomarkers sIgG, sIgM, sIgA, and FER obtaining the model in Equation (1):(1)$$\hat{\mathrm{IgT}}=-0.20+0.76\;\left[\mathrm{IgG}\right]+1.02\;\left[\mathrm{IgM}\right]+1.91\;\left[\mathrm{IgA}\right]+0.43\;\left[\mathrm{FER}\right]$$

It can be observed that the higher the concentration of IgG, IgM, IgA, and FER, the higher the concentration of sIgT. In this case, the goodness of fit found was 70.29% (Adjusted R-Squared). This means that the total level of specific immunoglobulins against SARS-CoV-2 increases with increasing levels of sIgGs, sIgMs, and sIgGs, and FER is a specific biomarker for COVID-19.

Finally, given that we have classified the severity of COVID-19, (Severe, Moderate, and Mild) we considered studying these qualitative variables in the multiple regression model. This model can be observed in Equation (2), which in this case, explains the total amount of specific immunoglobulins against SARS-CoV-2 with a goodness of fit of 73.01% (Adjusted R-Squared).
(2)$$\hat{IgT}=22.72+0.78\;\left[\mathrm{IgG}\right]+1.05\;\left[\mathrm{IgM}\right]+0.88\;\left[\mathrm{IgA}\right]+0.41\;\left[\mathrm{FER}\right]-58.92\;\left[\mathrm{Mild}\right]-14.78\;\left[\mathrm{Moderate}\right]$$

In this model, Severe patients corresponded with the intercept, which had a coefficient of 22.72. It was clearly observed that the higher the sIgG, sIgM, sIgA, and FER, the higher the level of IgT. Moreover, it is worthy to note that Mild and Moderate patients exhibited decreased levels of sIgT, which is quite sensible, and this result could suggest that the severity of COVD19 is related to the total amount of sIgTs and FER.

## 4. Discussion

In this work, we report the benefits of using this type of in vitro diagnostic system, which consists of a highly multiplexing diagnostic KIT based on IODM and ΔIROP (%) measurement to evaluate the COVID-19 disease, considering multiple biomarkers by a rapid, cost-effective, affordable, and reliable method. In addition, its handling is perfectly compatible with the usual practice in clinical analysis laboratories.

For this particular study, we considered six biomarkers: sIgT, the aforementioned specific SARS-CoV-2 antibodies, and two inflammatory biomarkers. Regarding the qualitative analyses in serum, a higher level of sIgT was observed in Severe patients compared to Moderate and Mild patients. This was the case, as well, for the biomarkers sIgG and sIgM. We also found differences in the inflammation marker FER, which was higher in patients. However, no significant differences were found for the other marker CRP. It is relevant to stress that at the end of February 2020, over 25% of blood donors at Hospital Clínico San Carlos showed previous contact with SARS-CoV-2.

We evaluated the biomarkers considered and built a multiple regression model to analyze the influence of these biomarkers on COVID-19. As a result, it is worth mentioning that this model explained the influence of the sIgG, sIgM, sIgA, and FER in the observation of sIgT with a goodness of fit of 73.01% (Adjusted R-Squared).

It must be considered that the sIgGs, sIgMs, and sIgAs biomarkers were obtained in a second incubation step with a secondary antibody, in contrast with sIgT, which was obtained directly due to the change in the interferometry pattern (label-free detection) after the first incubation step. Therefore, it is logical that there was no direct correction. However, even considering that the detection mechanism was different, it is remarkable that the results obtained are highly consistent, and it can be observed that the higher the sIgGs, sIgMs, and sIgAs, the higher the level of sIgT. It is also relevant to consider how the FER biomarker behaves as a specific biomarker for COVID-19, in contrast with the unspecific CRP biomarker in the multiple regression model obtained. Finally, we must emphasize that when the severity of the disease was included in the multiple regression model, the patients with higher levels of specific immunoglobulins against SARS-CoV-2 corresponded to the Severe patients, suggesting that the severity of COVD19 is related to the total amounts of sIgTs and FER.

Finally, in this work, we report a case study for measuring six biomarkers exhibiting relevant results and demonstrating the capacity for this technology to be transferred and used in clinical practice. It is also worth mentioning here that this technology can be easily designed to measure multiple biomarkers in order to improve the screening capacity in the future. With the ability to simultaneously measure many biomarkers, for example, this KIT can be designed for other relevant biomarkers as prognostic factors of the disease, such as some cytokines, in a single patient sample, while at the same time justifying and validating the benefits of using highly multiplexed KITs to predict disease and to investigate which are the most appropriate biomarkers, speeding up the screening capacity in an affordable way.

## Figures and Tables

**Figure 1 biosensors-12-00671-f001:**
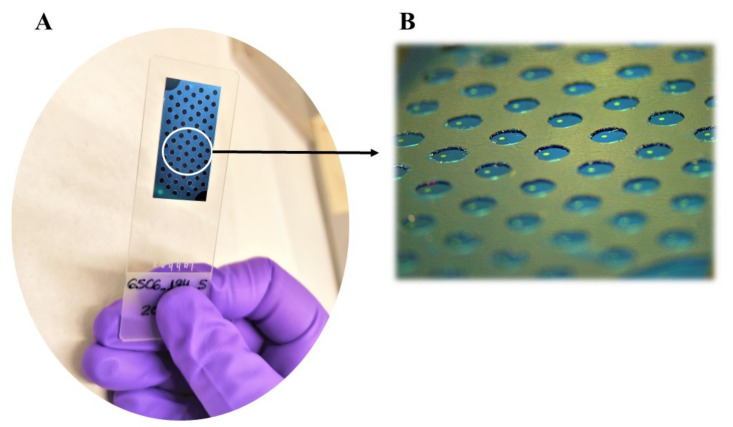
(**A**) Multiplexed 65-well KIT used for the described assays. (**B**) Detail of the sensor used, highlighting the BICELLs in yellow surrounded by the PVC. Abbreviations: Biophotonic Sensing Cell (BICELL), Polyvinyl Chloride (PVC).

**Figure 2 biosensors-12-00671-f002:**
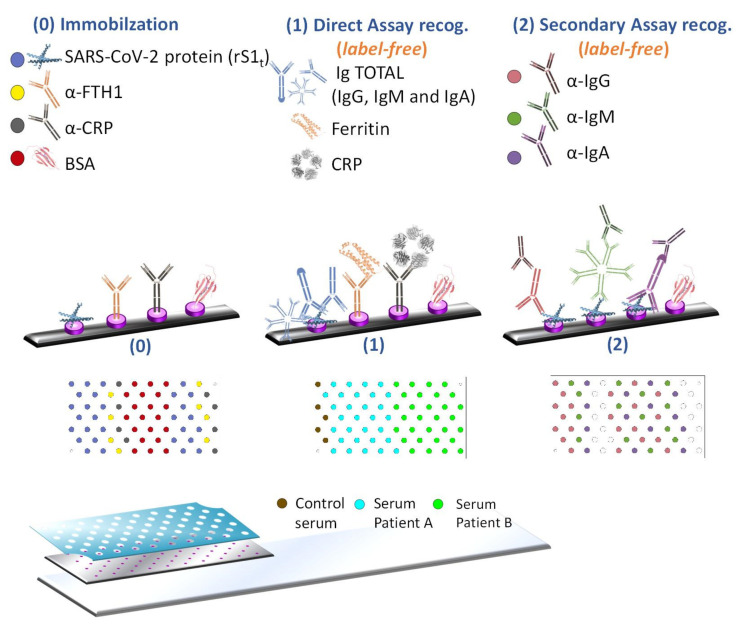
Depiction of the methodology followed for SARS-CoV-2 detection. The diagnostic kit used for the multiplexed diagnosis (on the bottom of the figure) is composed of 65 biosensing sites (purple dots represented in the picture) made of SU-8 epoxy resist and fabricated on a Silicon substrate chip placed on a glass slide to facilitate its handling. The upper layer is a PVC container (a blue sheet on the image), which holds/separates the micro-volume samples. To carry out the assay, as a first step (0), the immobilization of specific bioreceptors is performed. Later, in step (1), the direct recognition of SARS-CoV-2 specific biomarkers sIgTs, FER, and CRP is measured from three different types of serum samples (two samples from unknown patients, and a control serum). Finally, step (2) determines the biosensing signal of sIgG, sIgM, and sIgA through incubation with secondary antibodies. Abbreviations: Severe Acute Respiratory Syndrome Coronavirus 2 (SARS-CoV-2), Anti-Ferritin heavy chain (α-FTH1) and anti-C Reactive Protein (α-CRP), recombinant SARS-CoV-2 spike protein (rS1), Bovine Serum Albumin (BSA) and C Reactive Protein (CRP).

**Figure 3 biosensors-12-00671-f003:**
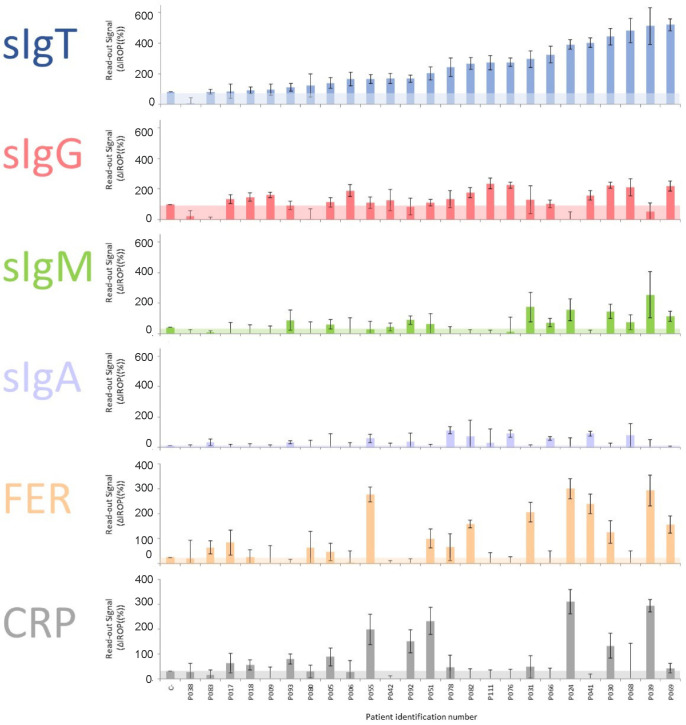
Results of the biomarkers measured in samples from patients with Severe symptoms. The cut-off for each marker is shown in colored shading. Abbreviations: total specific Immunoglobulins (sIgT), specific Immunoglobulins G (sIgG), specific Immunoglobulins M (sIgM), specific Immunoglobulins A (sIgA), together with Ferritin (FER), Reactive Protein (CRP) and ΔIROP (%) (Increased Relative Optical Power.

**Figure 4 biosensors-12-00671-f004:**
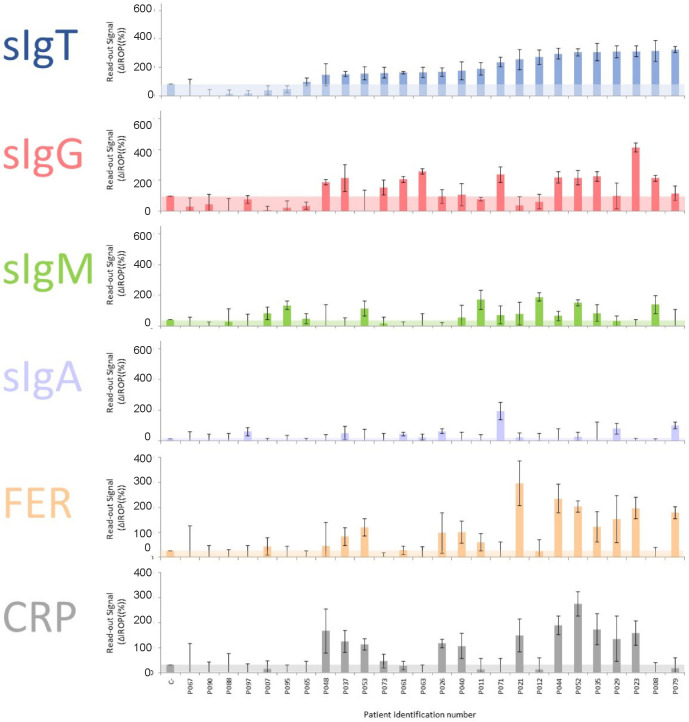
Results of the biomarkers measured in samples from patients with Moderate symptoms. The cut-off for each marker is shown in colored shading. Abbreviations: total specific Immunoglobulins (sIgT), specific Immunoglobulins G (sIgG), specific Immunoglobulins M (sIgM), specific Immunoglobulins A (sIgA), together with Ferritin (FER), C Reactive Protein (CRP) and ΔIROP (%) (Increased Relative Optical Power.

**Figure 5 biosensors-12-00671-f005:**
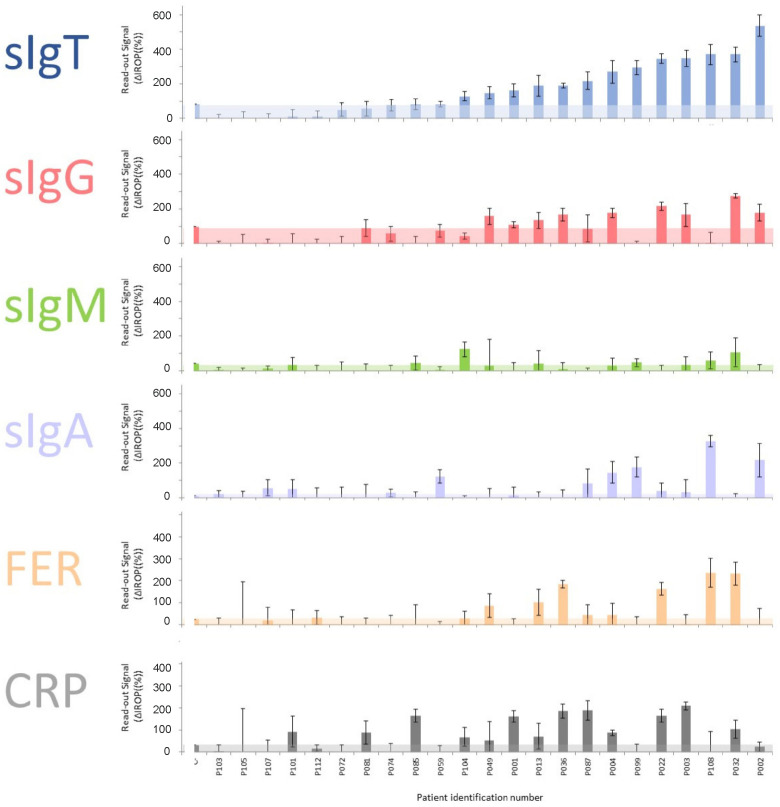
Results of the biomarkers measured in samples from patients with Mild symptoms. The cut-off for each marker is shown in colored shading. Abbreviations: total specific Immunoglobulins (sIgT), specific Immunoglobulins G (sIgG), specific Immunoglobulins M (sIgM), specific Immunoglobulins A (sIgA), together with Ferritin (FER), Reactive Protein (CRP) and ΔIROP (%) (Increased Relative Optical Power.

**Figure 6 biosensors-12-00671-f006:**
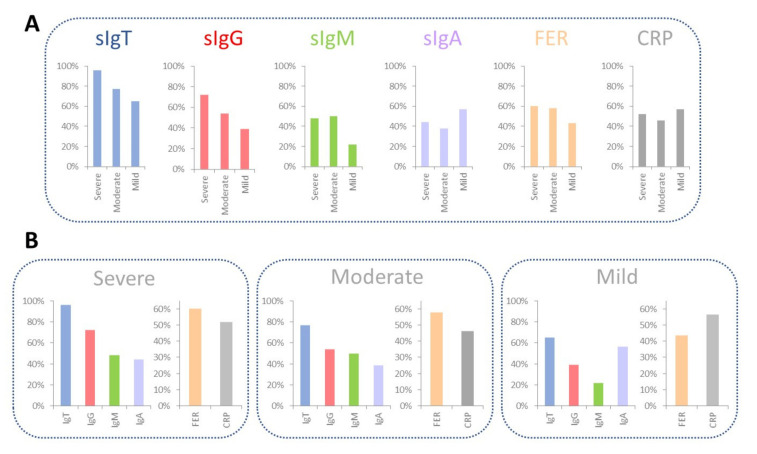
Summary of the qualitative comparison. (**A**) Percentage of positives for each biomarker as a function of the LSC. (**B**) Classification of each LSC as a function of the titers of sIgT, sIgG, sIgM, sIgA, FER, and CRP. Abbreviations: Level of Severity of COVID-19 (LSC), total specific Immunoglobulins (sIgT), specific Immunoglobulins G (sIgG), specific Immunoglobulins M (sIgM), specific Immunoglobulins A (sIgA), together with Ferritin (FER), C Reactive Protein (CRP) and ΔIROP (%) (Increased Relative Optical Power.

**Figure 7 biosensors-12-00671-f007:**
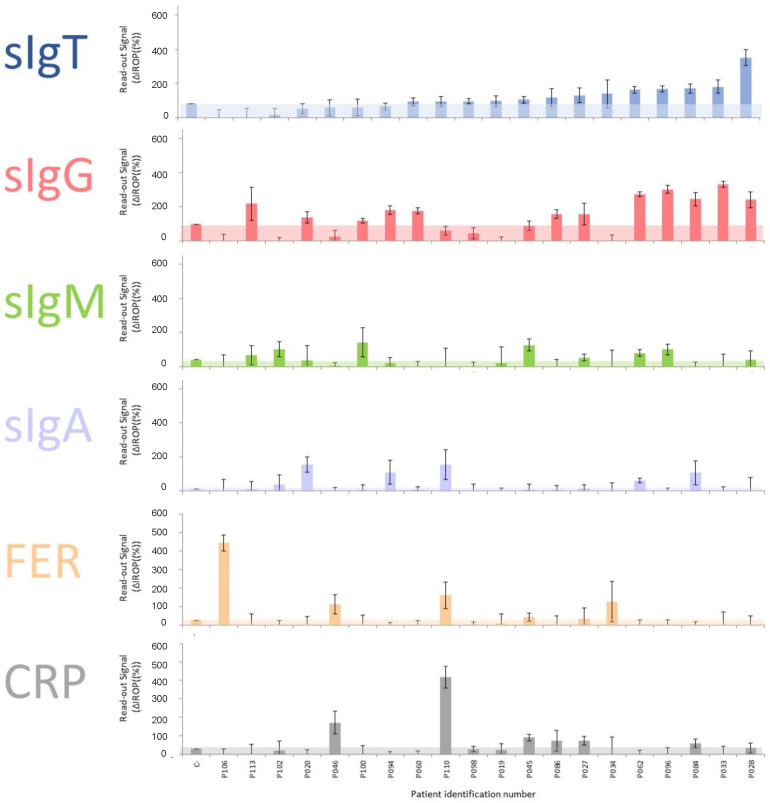
Asymptomatic cases of blood donors at the end of February 2020. Abbreviations: total specific Immunoglobulins (sIgT), specific Immunoglobulins G (sIgG), specific Immunoglobulins M (sIgM), specific Immunoglobulins A (sIgA), together with Ferritin (FER), C Reactive Protein (CRP) and ΔIROP (%) (Increased Relative Optical Power.

**Figure 8 biosensors-12-00671-f008:**
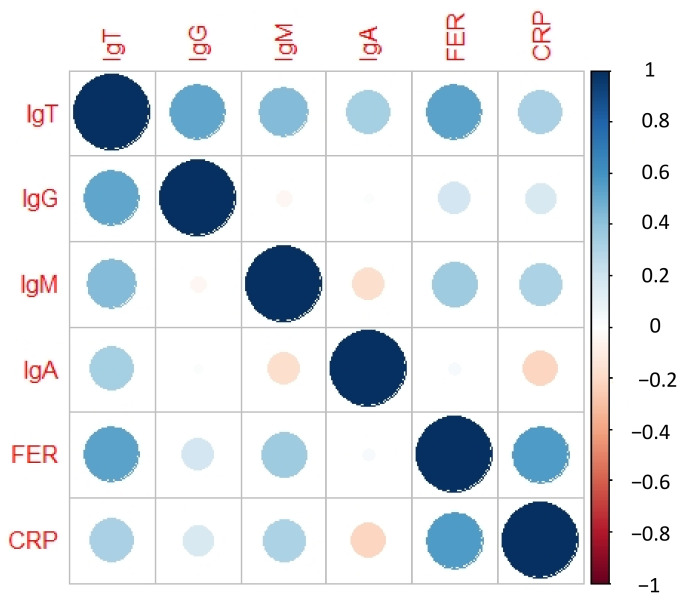
Correlation matrix. The size of the circles shows the correlation level highlighted on the left bar of the figure. Abbreviations: total specific Immunoglobulins (sIgT), specific Immunoglobulins G (sIgG), specific Immunoglobulins M (sIgM), specific Immunoglobulins A (sIgA), together with Ferritin (FER) and C Reactive Protein (CRP).

**Table 1 biosensors-12-00671-t001:** *p*-values obtained for the model of the simple linear regression model. *p*-values lower than the statistically significant level of (α = 0.05) can be considered to be related to sIgT.

	sIgG	sIgM	sIgA	FER	CRP
sIgT	1.59 × 10^−6^	0.000122	0.00362	1.17 × 10^−6^	0.00362

## Data Availability

Not applicable.

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
