# Peer review of "Integration of Multiple Interferometers in Highly Multiplexed Diagnostic KITs to Evaluate Several Biomarkers of COVID-19 in Serum"

_biosensors, 2022, doi:10.3390/bios12090671_

Round 1

Reviewer 1 Report

The manuscript by M. Holgado et al developed highly multiplexed diagnostic KITs based on an interferometric optical detection method developed in their previous work. They evaluated six COVID-19related biomarkers including sIgT, sIgG, sIgM, sIgA, FER and CRP. The results are rich and detailed, the discussion is specific, and will have a wide range of applicability and readers after addressing the following question:

 In Figure 6, the authors are not in-depth enough to discuss the comparison between the three groups including severe, Moderate and mild. It is recommended that the author add the quantification of differences between multiple groups, such as obtaining the corresponding p-value through ANOVA comparison, which is helpful for readers to understand the differences between different groups. Also, the error bars are necessary for Figure 6.

Reviewer 2 Report

The authors present an interesting work with a high potential for application.

However, some inconsistencies need to be reviewed and corrected. Writing needs improvement.

Major revisions

*The title mentions "integration of multiple interferometers" and multiple biomarkers. In my opinion, the title should be adjusted, it may raise doubts, and in addition, it is repeated "multiple". Authors may decide to choose where to apply the word.

*L25-26: I suggest that the authors rewrite the following sentence: to simultaneously measure different types of specific Severe Acute Respiratory Syndrome Coronavirus 2 (SARS-CoV-2) antibodies, as well as two inflammatory biomarkers and also include the specific biomarkers in this sentence and not divide them in another sentence.

*L32-33: Regarding the error obtained, how should it be expressed, as a percentage?

* L103: The authors refer that 74 positive samples were analysed. What about negative samples to be used as a control?

* L126: The authors refer that 94 samples were analysed. This information is not in line with the previous one. What about the control samples (negative results)?

* L165: Bioreceptors were immobilized through covalent binding. How? Which reagents were used? How long does the reaction take place? L170: Which proteins, the bioreceptors?

*L169: Why BSA is used as negative control and does not use the serum of a negative patient?

*L188: In serum samples from two patients, what kind of patients, negative COVID-19 patients? Do you mean 2 negative controls and 1 positive control?

*L197-203: Why is the sample signal subtracted from the BSA signal? Shouldn't certified human serum be used so that the matrix is uniform and thus perform a sign subtraction with the same matrix?

*L238: Authors mention Figure 8 which is the last Figure. Shouldnt it be inserted as Figure 3 since it corresponds to the initial data/results?

*General question to be verified in the full manuscript: How do authors calculate the residual standard error? Which is the unit, or percentage? Isn´t the medium value approximate to 70  a much higher error?

* At the end of the discussion part, in L421, the authors refer to the "the capability for this technology to be transferred and used in clinical practice". Please cite/refer to other works where this technology has already been reported.

Minor revisions

Introduction

L48-50: Authors mention a number of people, a percentage of children. This information is scarce, it is necessary to specify what it refers to, approximate percentages and data of the location to which it corresponds.

L52: Since several studies related to the pandemic have been carried out, please mention at what time of the pandemic the cohort was carried out.

L63-64: relevant is repeated. Despite being pertinent, avoid repeating the same words or expressions.

L74-75: In the sentence: it should be noted that high levels of interleukins have been reported, please mention in which conditions, namely the biological fluid maybe.

L79: authors refer difficult sample processing. In my opinion, in the case of the present work, the sample does not need to be processed. Maybe the sentence should be rewritten.

L82: The sentence seems incomplete. Do you mean but with simpler handling, low cost equipment and faster analysis?

L89: serums is incorrect. You can maybe use sera.

L103: The sentence should be rewritten, e.g., The present work reports the analysis of 74 serum samples

L184-186: Rewrite the sentence

Round 2

Reviewer 2 Report

*L32 and L265:The authors responded to the error percentage. But in the text, the exchange was not effected.

Why do you consider 71.92 as a residual standard error?

*In the introduction, paragraphs must be continuous, authors must remove the empty line between each one.

*Although I understand the issue of subtracting BSA, in my opinion the use of commercially obtained human serum is the most appropriate. The authors should reconsider this parameter in future work to obtain more reliable values.

*Figure 8 is very interesting and I understand the correlation. But in logical terms, it makes no sense to identify it on the L256 before the other figures. I suggest writing a reference that the data correlation is presented at the end of the document, but without mentioning the Figure in this part.
